# Cancer Cell-Intrinsic Alterations Associated with an Immunosuppressive Tumor Microenvironment and Resistance to Immunotherapy in Lung Cancer

**DOI:** 10.3390/cancers15123076

**Published:** 2023-06-06

**Authors:** Nerea Otegui, Maeva Houry, Imanol Arozarena, Diego Serrano, Esther Redin, Francisco Exposito, Sergio Leon, Karmele Valencia, Luis Montuenga, Alfonso Calvo

**Affiliations:** 1CCUN Cancer Center and Program in Solid Tumors, Center for Applied Medical Research (CIMA), University of Navarra, 31008 Pamplona, Spain; notegui@alumni.unav.es (N.O.); mhoury@alumni.unav.es (M.H.); dserrano@unav.es (D.S.); sleon.1@alumni.unav.es (S.L.); kvalencia@unav.edu (K.V.); lmontuenga@unav.es (L.M.); 2Department of Pathology, Anatomy and Physiology, School of Medicine, University of Navarra, 31008 Pamplona, Spain; 3Instituto de Investigación Sanitaria de Navarra (IDISNA), 31008 Pamplona, Spain; imanol.arozarena.martinicorena@navarra.es; 4Cancer Signaling Unit, Navarrabiomed, University Hospital of Navarra (HUN), Public University of Navarra (UPNA), 31008 Pamplona, Spain; 5Department of Medicine, Memorial Sloan Kettering Cancer Center, New York, NY 10065, USA; redinre@mskcc.org; 6Yale Cancer Center, New Haven, CT 06519, USA; francisco.exposito@yale.edu; 7Department of Pathology, Yale School of Medicine, New Haven, CT 06510, USA; 8Centro de Investigación Biomédica en Red de Cáncer (CIBERONC), ISCIII, 28029 Madrid, Spain

**Keywords:** lung cancer, immunotherapy resistance, cancer cell-intrinsic, gene mutations

## Abstract

**Simple Summary:**

Immunotherapy for non-small cell lung cancer (NSCLC) is a clinical reality with impressive efficacy for some patients. However, less than half of them will benefit from this therapeutic regime, and finding indicators of response is necessary for selecting the patients likely to respond. So far, detection of PD-L1 in tumors by immunohistochemistry is the only validated predictive biomarker. In recent years, certain changes in the tumor cells (intrinsic alterations), including mutations, epigenetic changes and metabolic rewiring, have been shown to modify the type of tumor microenvironment (TME) where such a tumor grows. This TME may determine response or refractoriness to immunotherapy. Examples of key mutations are *KRAS*, *SKT11*(*LKB1*), *KEAP1* and *TP53*, as well as co-mutations of these genes. Reshaping the TME with novel strategies in these particular TMEs could increase the efficacy of immunotherapy in lung cancer patients.

**Abstract:**

Despite the great clinical success of immunotherapy in lung cancer patients, only a small percentage of them (<40%) will benefit from this therapy alone or combined with other strategies. Cancer cell-intrinsic and cell-extrinsic mechanisms have been associated with a lack of response to immunotherapy. The present study is focused on cancer cell-intrinsic genetic, epigenetic, transcriptomic and metabolic alterations that reshape the tumor microenvironment (TME) and determine response or refractoriness to immune checkpoint inhibitors (ICIs). Mutations in *KRAS*, *SKT11*(*LKB1*), *KEAP1* and *TP53* and co-mutations of these genes are the main determinants of ICI response in non-small-cell lung cancer (NSCLC) patients. Recent insights into metabolic changes in cancer cells that impose restrictions on cytotoxic T cells and the efficacy of ICIs indicate that targeting such metabolic restrictions may favor therapeutic responses. Other emerging pathways for therapeutic interventions include epigenetic modulators and DNA damage repair (DDR) pathways, especially in small-cell lung cancer (SCLC). Therefore, the many potential pathways for enhancing the effect of ICIs suggest that, in a few years, we will have much more personalized medicine for lung cancer patients treated with immunotherapy. Such strategies could include vaccines and chimeric antigen receptor (CAR) cells.

## 1. Introduction

Lung cancer exhibits the highest mortality rates of all cancers and is the second most diagnosed cancer type worldwide, with 2.2 million new cases (1 in 10, 11.4%) and 1.8 million deaths (1 in 5, 18%) in 2020 [1]. Non-small-cell lung cancer (NSCLC) represents 85% of all lung cancer cases and is classified into three different histological subtypes: adenocarcinoma (LUAD; 40% of all lung cancers); lung squamous cell carcinoma (LUSC; 25%) and large cell carcinoma (LCC; 6%) [2]. Small-cell lung cancer (SCLC), which accounts for 15% of all lung cancer cases, is a highly aggressive type, with patients showing the lowest overall survival (OS): the median survival is lower than 2 years for early-stage patients and only 1 year for metastatic patients.

Genetic alterations found in NSCLC oncogenes include the following: *EGFR* (27% LUAD, <8% LUSC); *KRAS* (32% LUAD, 3% LUSC); *MET* (7% LUAD); *ALK* (<8% LUAD); *BRAF* (7% LUAD, 4% LUSC); *RET* (1% LUAD); *ROS* (2% LUAD), *PIK3CA* (4% LUAD, 16% LUSC) and *ERBB2/3* (3% LUAD, 4% LUSC). Frequent genetic alterations in tumor suppressor genes (TSG) are as follows: *TP53* (46% LUAD, 90% LUSC); *STK11* (17% LUAD, 2% LUSC); *KEAP1* (19% LUAD, 12% LUSC); *PTEN* (3% LUAD, 15% LUSC); *CDKN2A* (43% LUAD, 70% LUSC); *NF1* (11% LUAD, 11% LUSC) and *RB1* (7%LUAD, 7% LUSC) [3].

In the case of SCLC, recent “omics” technologies have produced a tremendous amount of valuable information about genetic alterations, novel SCLC subtypes and potential new targets. SCLC is characterized by *RB1* and *TP53* mutations in the majority of tumors [4]. In addition, inactivation of other genes of the *RB* family, such as p107 or p130, has been described [5]. SCLC also exhibits mutations in *PTEN* (10%), *CREBBP* (15–17%) and genes belonging to the Notch pathway (25%) [6,7,8]. Gene amplifications have been described for *MYC* family members (*MYC*, *MYCL*, *MYCN*) (20%) [9], *FGFR1* and *GNAS* [10,11]. 

The whole set of genetic alterations (total somatic mutation counts) in tumors configures the Tumor Mutation Burden (TMB), a surrogate biomarker for tumor neoantigen load. TMB has been shown in clinical trials to correlate with efficacy in lung cancer patients treated with immunotherapy [12]. However, there is currently no standard methodology to determine TMB. Lung cancer has one of the highest TMBs (approximately 8 mutations (mt)/megabase (Mb) on average), associated with the effect of tobacco smoke [13]. Genetic events that act as tumor drivers and TMB may shape the tumor microenvironment (TME) and make the tumor responsive or refractory to immunotherapy. Although the association between specific gene mutations and the architecture of the TME has been shown for some specific genes, we are just starting to understand how these driver mutations configure the stroma and immune cell populations that may affect response to immunotherapy.

In recent decades, the use of targeted therapy and, more recently, immunotherapy, has produced a spectacular improvement in clinical benefit, mainly in NSCLC [14]. Mutation analysis is currently used for several actionable targets, including *EGFR*, *KRAS, MET, HER2* and *BRAF*, as well as rearrangements in *ALK*, *RET* and *ROS*. The introduction of immune checkpoint inhibitors (ICIs), such as anti-PD-1, anti-PD-L1 and anti-CTLA-4, alone or in combination with chemotherapy, has been a breakthrough in oncology, with impressive clinical benefit in lung cancer and melanoma patients [15]. However, ICIs are only effective in less than half of lung cancer patients, and most responders will acquire resistance during the treatment course [16]. The mechanisms of resistance to immunotherapy have not been completely elucidated, and approaches aimed at overcoming them are necessary. Another problem is how to identify those patients who are likely to respond. Currently, only PD-L1 tumor expression and TMB have been associated with ICI efficacy in NSCLC patients, although they are not accurate biomarkers [17]. Many other potential biomarkers have been identified, but they have not been validated yet. Several other proteins that act as immunological checkpoints have been discovered, including TIM-3, LAG-3, TIGIT, VISTA and Siglec-15 [17]. They could serve as targets in novel immunotherapy strategies and as predictive biomarkers, but these new options are still under clinical investigation. 

During tumor evolution, cancer cells adapt towards the development of immune escape mechanisms that increase tumor heterogeneity, allow the tumor to grow and spread and contribute to resistance to therapy [18]. LUAD and LUSC preinvasive lesions are characterized by reduced expression of MHC-I and diminished antigen presentation, a decreased number of effector T cells, activation of several immune checkpoints and the secretion of IL-10 and IL-6 (immunosuppressive cytokines) [19,20]. There is discussion about the convenience of continuing ICI treatment once progression is observed in patients, because an initial pseudoprogression may precede an objective response. However, pseudoprogression is uncommon (<10% of NSCLC patients), and ICI continuation should only be considered in patients with clinical benefit and no severe side effects [21]. Personalized treatments in this situation would require knowing the post-treatment TME through a biopsy, something that is not routinely performed. The ICI-resistant TME could express other immune checkpoints that have been associated with acquired resistance for which therapeutic antibodies are available, including anti-LAG-3, anti-TIM-3 and anti-TIGIT [22]. Apart from ICIs, other therapeutic strategies using anti-VEGF, radiotherapy, STING agonists, IDO inhibitors or FAP-blocking agents could overcome acquired resistance [22].

Evidence from preclinical studies and clinical trials has shown that co-occurring mutations in driver oncogenes and TSG are important contributors to the heterogeneous responses seen in the clinic [23,24,25,26]. The crosstalk between cancer cells and the TME is determined not only by genetic alterations, but also by epigenetic, transcriptomic, proteomic and metabolomic changes in the cancer cell. To really understand the specific TME of a tumor, all these factors would need to be considered to determine either response or resistance to ICIs. However, as this will not be clinically feasible, identification of a few key markers (either mutations, cytokines or in situ expression of certain proteins) associated with each TME would help in decision making about the use or not of immunotherapy. 

In this review, we address the main genetic, epigenetic and metabolic changes associated with response to ICIs in lung cancer, including NSCLC and SCLC. We also describe new emerging pathways linked to the establishment of particular TMEs that can be favorable or unfavorable for immunotherapy response.

## 2. Main Gene Mutations That Shape the Tumor Microenvironment in NSCLC

### 2.1. KRAS Mutations

Oncogenic mutations in *KRAS* are found in ~30% LUADs. *KRAS* mutant tumors are characterized by their ability to evade immunosurveillance through different mechanisms, including the secretion of immunosuppressive inflammatory cytokines (such as IL-6, IL-1b and GM-CSF) and activation of NF-kB and STAT3 signaling pathways [27]. These tumors also express higher levels of PD-L1 and recruit myeloid-derived suppressor cells (MDSCs), regulatory T cells (Tregs) and “M2-like” macrophages in the TME [27,28]. In spite of this immunosuppressive TME, patients with *KRAS* mutant tumors tend to respond better to ICIS than those with other oncogenic driver alterations such as *EGFR*, *BRAF*, *MET*, *HER2*, *ALK*, *RET* and ROS1 [29]. 

The different *KRAS* mutation subtypes might impact outcomes differently in ICI-treated patients, although results are still inconclusive. Jeason et al. studied the efficacy of ICIs in tumor samples from NSCLC patients harboring G12A, G12C, G12D, G12V or G13C *KRAS* mutations, and found no differences in progression-free survival (PFS) or OS between these major *KRAS* mutation subtypes [30]. Similarly, another study showed that the outcomes of patients with G12C *KRAS* mutations did not differ from those with any other *KRAS* alteration [31]. In contrast, another study has shown that the *KRAS*-G12D mutation may drive immunosuppression and primary resistance to anti-PD-1/PD-L1 therapy [32]. Analysis of tumor samples from the aforementioned studies showed that, despite the apparent similar clinical benefit regardless of their *KRAS*-mutation subtype, patients with G12D, G12V or G13C mutations had a significantly higher PD-L1 expression than those with G12A or G12C mutations [30]. Taken together, these data highlight the need for future studies addressing how the different *KRAS*-oncogenic mutations alter tumor biology and how this may influence immunotherapy efficacy. 

The genetic heterogeneity of *KRAS* mutant tumors driven by co-occurring mutations with a variety of TSG, such as *STK11* (also called *LKB1*), *KEAP1* and *TP53*, is known to play a pivotal role in oncogenic transformation [33], TME modeling [34] and the modulation of the therapeutic responses to different treatments [35], including immunotherapy [23]. 

### 2.2. STK11 Mutations

*STK11* loss is present in nearly 17% of LUADs and in up to 30% of *KRAS* mutant tumors [33,36]. STK11, which directly phosphorylates and activates AMP-activated protein kinase (AMPK), exerts different functions related to the regulation of the cancer cell cycle, metabolism, angiogenesis, DNA damage response, epigenetic changes and cell differentiation [37]. Through repression of the NOX1/VEGF axis and activation of mTOR and HIF1-α, *STK11* loss triggers tumor vascularization [38,39]. In parallel, *STK11* inactivation facilitates tumor growth by inducing an immunosuppressive TME, a process where STING (STimulator of INterferon Genes) plays a key role [40,41] (Figure 1). In non-malignant tissues, STK11 potentiates STING signaling to detect double-strand genomic and mitochondrial DNA. This leads to increased expression of type I interferons and inflammatory cytokines such as CCL3 and CCL5, which work in an autocrine/paracrine fashion to activate antigen-presenting dendritic cells (DCs) (both cross-priming T cells and MHC-I) [42,43,44]. In addition, the STK11/STING axis regulates the expression of PD-L1 [40]. Consequently, *STK11* deficiency triggers the opposite effect, characterized by reduced tumor infiltrating lymphocytes (TILs) and decreased tumor PD-L1 expression. Furthermore, a lack of STK11 signaling triggers the accumulation of pro-tumoral immune cells caused by tumor expression of IL-6, CXCL7, CXCL5 or G-CSF, thus favoring the recruitment of neutrophils [42]. In turn, high expression of arginase-1 (ARG-1) and IL-10 in tumor-associated neutrophils can trigger Treg cell expansion and T cell exhaustion [36]. 

Regulation of the STING pathway by inactivation of *STK11* can occur through different epigenetic mechanisms. First directly, by preventing AMPK-dependent inhibitory phosphorylation of EZH2 and DNMT1 [45]. Indirectly, *STK11* loss and subsequent AMPK inactivation increase intracellular levels of S-adenosyl methionine (the substrate of the methyltransferase DNMT1), which, in turn, trigger the activity of EZH2 and DNMT1 to block STING expression by promoter methylation [41]. This switches off the expression of inflammatory cytokines and establishes the immunosuppressive TME. Increased expression of *MYC* has also been observed in *STK11*-mutated tumors, which may contribute to the immune-inert phenotype due to the IL-23- and CCL9-mediated exclusion of B, T and natural killer (NK) cells [46].

*KRAS/STK11* double mutant tumors are particularly aggressive and refractory to ICIs due to the immunosuppressive features of their TME: low PD-L1 expression, fewer TILs and an increased number of Tregs [25,33,42]. These tumors suppress antigen processing through compromised immunoproteasome activity and increased autophagic flux [47]. In *KRAS/STK11* tumors, STING is silenced, resulting in protection from STAT1-induced cytotoxicity and induction of NF-kB-mediated secretion of pro-tumorigenic cytokines such as IL-6 [41,48].

### 2.3. TP53 Mutations

*TP53* mutations are found in >50% of lung cancers [49]. It has been widely demonstrated that *TP53* loss of function increases genomic instability and generates DNA damage, which links *TP53* deficiency with higher TMB and correlates it with clinical benefit in advanced NSCLC patients treated with ICIs [50]. Moreover, the number of CD8+ TILs is increased in *TP53*-mutated tumors, but only *TP53* missense mutations are associated with high PD-L1 expression [51]. JAK-STAT pathway enrichment and IFNγ signatures are also present in missense-mutated *TP53* tumors [52]. Suppressor immune cells, such as M2 macrophages and neutrophils, are expanded in *TP53* nonsense mutated tumors. *KRAS/TP53* co-mutated tumors show an inflammatory TME with abundant CD8+ lymphocytes and a higher TMB than *KRAS* or *TP53* single mutant tumors. Patients with *KRAS/TP53* co-mutations show improved PFS and OS [53]. 

Recent studies are showing that mutations in *TP53* can also promote a gain-of-function (GOF) related to immunosuppression. In this regard, TP53 has been shown to play a key role in the regulation of epithelial-to-mesenchymal transition (EMT), a process that is linked to the expression of immune checkpoints via chemokine production, leading to immune evasion [54]. Classical EMT is characterized by the acquisition of mesenchymal properties, motility and cancer invasion, but EMT is also associated with multiple other molecular and cellular events, such as the recruitment of MDSC and the expression of PD-L1 [54]. The GOF mutation p53-R175H has been shown to upregulate the EMT inducer TWIST in cancer cells [55]. Many other intracellular mechanisms are involved in p53 GOF mutations and EMT, involving miRNAs, KLF17, FOXM1, ZNF652, etc. [56]. The acquisition of EMT features in NSCLC has been associated with lower susceptibility to cytotoxic T cells (CTL) and natural killer (NK) cells, an increase in M2-like macrophages, MDSC and Tregs, and the release of immunosuppressive cytokines such as TGF-β, TNF-α, IL-10, and arginase-1 [57].

### 2.4. EGFR Mutations

Activating mutations of *EGFR* are found in 10–30% of NSCLC patients, most of them in LUADs. The clinical benefit of PD-1/PD-L1 in patients with *EGFR*-mutated tumors is minimal, but the type of mutation seems to have a different impact on the outcome: the presence of the L858R mutation, exon 19 deletions or the T790M resistant mutation seem to play a role in the response to ICIs [58,59]. Immunosuppressive effects of *EGFR* mutations involve an increase in Tregs, MDSCs and tumor-associated macrophages (TAMs), as well as lower numbers of CD8+ T cells [60,61]. Tumor infiltration of Tregs is favored by indoleamine 2, 3-dioxygenase (IDO) production by DCs, which promotes Tregs conversion [59]. In *EGFR*-defective tumors, an upregulation of immunoglobulin-like transcript 4 (ILT4) has been described. This leads to increased M2 polarization and impairment of T-cell proliferation and cytotoxicity [60,62]. ILT4 inhibition promoted anti-PD-L1 efficacy, followed by a decrease in TAMs and Tregs [60,62]. EGFR-Tyrosine Kinase Inhibitors (TKIs) may synergize with ICIs due to the upregulation of MHC classes I and II, suppression of Tregs and decrease in PD-L1 expression [60]. The combination of TKIs with immunotherapies is being tested in phase I trials and includes the use of Nivolumab (NCT01454102), Pembrolizumab (NCT02039674) and Atezolizumab (NCT02013219) [63].

### 2.5. KEAP1 Mutations

Mutations in the gene coding for Kelch-like ECH-associated protein 1 (KEAP1) have also been linked to de novo resistance to ICIs [64,65]. *KEAP1* loss occurs in a substantial proportion of lung cancer patients (19% LUADs and 12% LUSCs) and is associated with an immunosuppressive TME characterized by low infiltration of CD8+ TILs and NK cells [66,67]. The functions of KEAP1, a negative regulator of NFE2L2/NRF2 with a main role in regulating the oxidative damage response, are discussed in the next section of this review. LUAD patients with tumors deficient in *KEAP1* and mutations in either *KRAS* or *STK11* showed inferior OS after ICIs, despite having high TMB [68]. Complex combinations of oncogenic drivers and TSG mutations, such as *KRAS/TP53/KEAP1,* also contribute negatively to ICI response rates compared to *KEAP* wild-type (WT) tumors [27,67]. 

### 2.6. Mutations in Genes of the Antigen Processing and Presentation Machinery

Loss of MHC-I and a lack of antigen presentation are other common mechanisms utilized by tumors to escape the cytotoxic effect of CD8+ T cells [69]. In addition, impaired INF-γ signaling can also synergize with loss of antigenicity and reduce the functionality of the immune system [70,71,72]. The cancer cell-immune cell interaction evolves as the tumor progresses and has been subdivided into three phases: (a) elimination, (b) equilibrium, and (c) escape [73,74]. At the beginning of tumor development, the immune system very effectively eliminates newly formed and highly antigenic cancer cells. If the immune system cannot eradicate all cancer cells, it controls tumor growth but cannot fully get rid of it (the equilibrium phase). It is in this phase that immune evasion can start taking place, either by selecting cancer cells that are less immunogenic, or by inducing an immunosuppressive milieu. The escape phase consists of success in cancer growth by evading the immune system. In this process, the lack of antigen presentation confers a clear advantage for immunosuppression. In fact, there is a relationship between mechanisms of resistance to ICIs and this process of immunoediting [75]. Antigen presentation by cancer cells depends on the correct activity of multiple proteins that belong to the antigen processing and presentation (APP) complex [76]. Therefore, alterations in the components of the APP complex are frequently found in tumors, thus avoiding immune recognition [77]. These defects can result not only in the lack of cell membrane-bound MHC-I but also in the type of peptides presented to the T lymphocytes. All this leads to impaired antitumor responses and resistance to ICIs [78].

MHC-I is a heterodimeric complex consisting of a polymorphic heavy chain and an invariable light chain named β-2-microglobulin (B2M). Genetic alterations in *B2M*, loss of heterozygosity or downregulation lead to MHC-I instability and impaired antigen presentation [79]. Around five percent of tumors from untreated NSCLC patients carry somatic mutations in *B2M*, which is associated with a lower number of CD8+ lymphocytes [80]. Other genes of the APP (i.e., *CALR*, *PDIA3*, and *TAP1*) implicated in the maturation of HLA-I, are also altered in NSCLC [80]. Interestingly, restitution of B2M expression in lung cancer cells upregulates targets of IFN-α/IFN-γ pathways [80]. *B2M* mutations have also been found in NSCLC patients as an acquired mechanism of ICI resistance [81]. *B2m* knockout using CRISPR technologies in an immunocompetent LUSC mouse model caused resistance to anti-PD-1 [81].

### 2.7. Mutations in other Genes with Potential Importance in Immunosuppression

The extent to which other genetic alterations contribute to primary or acquired resistance to immunotherapy in NSCLC still needs further investigation. Another possible candidate TSG is the phosphatase and tensin homolog (PTEN), whose loss of function has been previously reported to confer resistance in melanoma, uterine leiomyosarcoma and prostate cancer [82,83,84]. Although there is still limited data on the contribution of *PTEN* loss to immunotherapy resistance in NSCLC, one study described that *PTEN* mutations were only found in patients who did not respond [85]. A recent case report has shown that a *PTEN* mutation was found upon treatment with Nivolumab in a LUAD patient harboring an *ERBB2*-driver mutation [86]. Another study using animal models of NSCLC has shown that co-deletion of *Keap1* and *Pten* resulted in immunologically cold tumors with increased PD-L1 expression [66]. However, further studies are needed to assess whether *PTEN* mutations alter the TME and if this is involved in the response to ICIs.

Notch signaling regulates tumor angiogenesis and activates cytotoxic T cells, promoting the maturation of naive CD8+ T cells, the secretion of IFN-γ and the polarization of macrophages towards the M1 phenotype [87]. Li et al. found that NSCLC patients with high mutation rates in the Notch signaling pathway had significantly improved PFS and OS compared to those without that signature. The highly mutated Notch signature was related to an inflammatory and immunogenic TME [88].

Mutations in other genes have also been associated with a particular modification of the TME and, in some cases, with immunotherapy response/refractoriness. Those include *ALK*, *ROS1*, *ZFHX3*, *PTCH1*, *PAK7*, *UBE3A*, *TNF-α*, *LRP1B* and *FBXW7* [60]. However, more studies are needed to clarify their implication in ICI response.

## 3. Metabolic Rewiring in NSCLC and its Influence on the TME

The bioenergetic requirements of rapidly dividing cancer cells and immune cells imply a competition for essential nutrients, which has a profound effect on the cytotoxic activity of TILs. Tumor cells have a high demand for glucose, amino acids and lipids to cope with the demands of cell division. In the aerobic glycolysis (Warburg effect) that is characteristic of cancer cells, lactate is generated at high levels, thus creating an acidic microenvironment [89]. This has an effect on immune cells within the TME, favoring the presence of MDSC, M2-macrophages and Tregs, and reducing the ability of CD8+ TILs to kill tumor cells [89]. Glucose utilization by cancer cells metabolically limits the activity of T cells, as they are dependent on glycolysis. On the contrary, Tregs are able to differentiate and exert their immunosuppressive role in low glucose microenvironments, instead utilizing oxidative phosphorylation (OXPHOS) as an energy source [90]. 

### 3.1. Lactate

Lactate production by cancer cells has acquired increasing relevance in the context of TME as a byproduct of aerobic glycolysis. Tumor cells generate and excrete to the extracellular compartment a high amount of lactate that drives tumor growth and metastasis [91,92]. Excretion of lactate requires the action of two monocarboxylate transporters, MCT1 and MCT4. Of note, expression of MCT4 but not MCT1 has been widely correlated with OS in a variety of cancers, including NSCLC [93,94]. Both transporters are controlled by CD147, a chaperone that regulates NSCLC tumor progression via increasing lactate efflux, reduction of extracellular pH and modulation of the immune microenvironment. Indeed, high glycolytic tumor activity and lactate production lead to reduced immune infiltration (e.g., CD3+ and CD8+ T cells), a process observed in NSCLC and across other tumor types [95]. At the molecular level, lactate triggers TGF-β signaling and PD-1 expression in Tregs [96,97]. Moreover, high extracellular lactate leads to increased PD-L1 expression in tumor cells [98]. Conversely, blockade of lactate production via inhibition of pyruvate dehydrogenase with oxamate led to increased CD8+ T cell infiltration, reduced NSCLC tumor growth and improved responses to anti-PD-1 antibodies [99]. In line with this, a recent report showed how a glucose-restricted diet led to reduced tumor growth and increased intratumoral infiltration of CD8 effector memory T cells and NK cells [100]. 

In parallel, increased extracellular lactate can lead to metabolic symbiosis, whereby low-glycolytic cells that rely on oxidative phosphorylation take advantage of available lactate to transport it into the cell, turn it into pyruvate, and use it as substrate for the tricarboxylic acid (TCA) cycle. As a result, these neighboring cells increase the levels of reduced forms of NAD+ that feed the electron transport chain to produce ATP. In fact, work by the DeBerdadinis group elegantly showed that lung tumors preferentially use lactate, rather than glucose, to fuel respiration [91]. Interestingly, lactate, uptaken by cells, can inhibit the enzyme prolylhydroxylase 2 (PHD2), leading to HIF-1α activation, angiogenesis and tumor growth [101]. Apart from MCT1 and 4, GPR81/HCAR1 acts as a receptor for lactate, regulating cancer survival [102].

### 3.2. Hypoxia

The metabolic reprogramming of cancer cells is amplified by the hypoxic conditions frequently found in tumors. Hypoxia orchestrates NSCLC progression and metastasis events, such as enrichment of a cancer stem cell (CSC) and epithelial-to-mesenchymal transition (EMT) phenotypes and immune escape [103]. Through the activity of HIF-1α, hypoxia reduces the migratory ability of effector T cells [104]. Luo et al. have shown that the HIF-1α inhibitor PX-478 (S-2-amino-3-[4′-N,N,-bis(chloroethyl)amino] phenyl propionic acid N-oxide dihydrochloride) in combination with anti-PD-1 causes marked tumor growth inhibition and prolonged survival in animal models of NSCLC, which correlates with increased TILs and granzyme B secretion [105].

### 3.3. Amino Acids

NSCLC also demands a high number of amino acids to fuel bioenergetic pathways. Some amino acids can be converted into acetyl-CoA, which will produce ATP through the TCA. Purine and pyrimidine biosynthesis, needed for DNA replication, is an amino acid-dependent process [106]. ROS production by cancer cells can damage the different macromolecules of the cell, leading to cell death. To inhibit ROS, cancer cells use different strategies, including the synthesis of glutathione from glycine, glutamate and cysteine, to regulate redox balance [106]. Serine contributes to the folate cycle, which generates a large amount of NADPH [107]. The metabolite kynurenine is generated from tryptophan by tryptophan 2,3-dioxygenase (TDO) and IDO, with a relevant function in cancer [108]. An increased proportion of kynurenine over tryptophan has been described in several cancer types, including NSCLC [109]. Kynurenine released from tumors causes CD8+ T cell death and promotes immune tolerance [110,111]. Tobacco smoke increases levels of the cysteine-glutamate antiporter SLC7A11 in NSCLC cells. By blocking this antiporter with sulfasalazine (SASP) in combination with anti-CTLA-4, Arensman et al. showed a dramatic increase in the frequency and durability of antitumor responses [112]. Byun et al. have shown that restricting glutamine metabolism in cancer cells results in higher PD-L1 levels in different cancer cells, including NSCLC [113]. Glutamine depletion together with anti-PD-L1 strongly promoted T cell cytotoxicity mediated by Fas/CD95 [113]. 

### 3.4. Lipids

Lipid metabolism is being acknowledged as a key factor in the control of tumor immunity [114]. Lipids are responsible for energy storage, the control of membrane fluidity and intracellular signaling. From the different types of lipids, cholesterol, fatty acids and prostaglandins have been linked to the regulation of cancer cell-intrinsic properties, such as proliferation, survival and metastasis [114]. Cancer cells can use fatty acid oxidation (FAO) and de novo lipid synthesis to produce more energy [115]. It has been described that changes in lipid composition can alter the role of immune cells within the TME and disrupt their antitumor activities [115]. For example, Shaikh et al. have shown that palmitic acid inhibits immune responses due to lowering MHC-I in the membrane, and Coutzac et al. have shown that high butyrate and propionate blood levels are associated with a lack of response to anti-CTLA-4 blockade and higher Treg infiltration [116,117]. Cholesterol in the membrane rafts increases T-cell signal transduction [118]. Prostaglandin E2 (PGE2) downregulates IL2 and INF-γ, inhibiting T-cell cytotoxicity [119]. Fatty acids promote the immunosuppressive function of MDSCs, and PGE2 increases the number of intratumor Tregs [120,121]. Moreover, fatty acids and PGE2 cause the release of IL-10 and TGF-β, which play an immunosuppressive role in the TME [122]. They can also induce the expression of immune checkpoints [114]. This exemplifies the importance of lipid metabolism in shaping the TME. 

As aberrant accumulation of lipids in cancer cells promotes an immunosuppressive TME, strategies to reduce the lipid content can improve antigen presentation and activation of cytotoxic T cells, thus reshaping the TME. Studies in animal models and clinical trials have tested the combination of therapies to normalize lipid metabolism and immunotherapy in cancer [114]. For example, PPARs are fatty acid sensors that regulate fatty acid oxidation, and the use of PPAR agonists has been proposed as an antitumor strategy that may enhance the immune system [123]. Inhibitors of the enzyme of hydroxyl methylglutaryl-coenzyme-A (HMG-CoA) to control cholesterol biosynthesis are statins, which potentiate the effect of immunotherapy in lung cancer [124].

### 3.5. Metabolic Rewiring by STK11 Alterations

*STK11* is a TSG operating at the heart of cell energy homeostasis. STK11 regulates glucose and lipid metabolism in response to energy fluctuations, represented by the relative intracellular levels of adenosine monophosphate (AMP) compared to adenosine triphosphate nucleotides (ATP). Upon reduced levels of ATP, STK11 binds to and phosphorylates AMPK, which, in turn, can phosphorylate and inactivate the TSC2 complex to block mTOR and stop cell proliferation [37]. At the same time, AMPK triggers lipid uptake and catabolism (e.g., fatty acid and cholesterol) as well as the expression of glucose transporters to increase the glycolytic flux and reverse energy deficiencies [125]. In non-malignant cells, *STK11* loss results in mitochondrial alteration, metabolic dysfunction and increased ROS levels that make cells unable to respond to metabolic stress [37] (Figure 1).

Concerning the alteration of metabolic pathways, some reports indicate that NSCLC cells mutant for *KRAS* and *STK11* become dependent on hexosamine biosynthesis and de novo fatty acid synthesis. Indeed, inhibition of either GFPT2 (glutamine-fructose-6-phosphate transaminase 2), a central enzyme of the hexosamine pathway, or the acetyl-coA carboxylase (ACAC, key for fatty acid synthesis), blocks NSCLC tumor growth [126,127]. Whether these metabolic features affect the immune TME remains to be elucidated. As the inactivation of AMPK by *STK11* mutations would likely lead to increased cholesterol biosynthesis, the use of statins as enhancers of ICI activity [124] might represent another link between STK11-mediated metabolic rewiring and the reshaping of the TME. As previously mentioned, *STK11* inactivation leads to increased levels of ROS and an intracellular redox imbalance [37]. As the tumor progresses, cancer cells can tolerate the presence of ROS better [128]. All in all, the poorer responses to ICIs in *STK11*-mutant NSCLC patients are probably the result of the cumulative effects described here, leading to the establishment of an immunosuppressive TME [25,129,130,131].

### 3.6. Metabolic Rewiring by KEAP1 Alterations

KEAP1 is the other metabolic-related protein that has been associated with a lack of response to anti-PD-1/PD-L1 antibodies. KEAP1 functions as an adaptor protein for Cullin3, an E3 ligase that negatively regulates NRF2 activity (Figure 1). KEAP1 binding to NRF2 directs the transcription factor towards degradation via the proteasome [132]. In situations of redox stress, elevated ROS modify cysteine residues on KEAP1, disrupting KEAP1/NRF2 binding and allowing NRF2 to translocate to the nucleus. This activates the transcription of genes involved in redox homeostasis and ROS detoxification [133]. *KEAP1* mutation allows NRF2 to promote the transcription of genes related to anti-redox activity (GCLC, xCT, GSR1; related to glutathion production), ROS detoxification (GPX2, GSTs, NQO1), NADPH regeneration (C6PD, PGD), the thioredoxin antioxidant system (TXN, TXNRD1) and iron metabolism (FTL, FTH) [134]. Therefore, *KEAP1* loss would facilitate the management of toxicity derived from high ROS levels, typical of a *STK11*-deficient background.

The KEAP1-NRF2 aberrant hyperactivation not only alters the redox balance of the cell but also modifies numerous metabolic-related pathways that, ultimately, affect the activity of immune cells within the TME. For example, it promotes cell proliferation and cancer growth by reprogramming glucose metabolism, favoring the use of TCA and the pentose phosphate pathway (PPP) [135]. NRF2 also upregulates glutaminase, thus converting glutamine to glutamate [136]. This could suggest a novel strategy against NSCLC based on cancer cells addicted to glutamate metabolism [135]. Several reports have shown that, in the context of lung cancer, inhibition of glutamine metabolism can have the capacity to reactivate CD8+ T cells, reduce the MDSC population and even promote the expression of PD-L1 [137,138]. Targeting glutamine metabolism would work as a double-edged sword by directly blocking tumor survival and by enhancing anti-tumor immune responses. Reports showing that glutamine metabolism inhibition can cooperate with ICIs in cancer cells provide the basis for future combinatorial treatments for NSCLC patients [113].

As we have previously discussed, *KEAP1* mutations in LUADs showed lower TILs as well as lower expression of genes of the antigen presentation machinery and inflammatory cytokines, compared to wild-type tumors [135]. In line with these observations, NRF2 has been described as an important regulator of the inflammatory response in the context of innate immunity. More precisely, NRF2 modulates the STING pathway by reducing STING mRNA stability [139]. In this regard, *KEAP1* mutations would further support the immunosuppressive TME induced by *STK11* loss (Figure 1), as suggested by clinical data. Intriguingly, STING can also regulate the expression of IDO, an enzyme that catabolizes the amino acid tryptophan in the pericellular compartment to produce N-formyl-kynurenine [140]. IDO can induce Lewis lung tumor growth through T cell inactivation, indicating that STING can promote tolerogenic responses [140]. NRF2 has been shown to regulate the kynurenine pathway to modulate tryptophan metabolism. Be it by activating *NRF2* mutations or *KEAP1* loss, NRF2 leads to increased expression of KYNU (tryptophan-kynurenine enzyme kynureninase), an enzyme of the tryptophan catabolic pathway acting downstream of IDO [141]. Accordingly, KYNU activity led to an immunosuppressive TME characterized by a higher proportion of Tregs and increased expression of PD-1 and PD-L1 [141].

In a *KRAS/STK11* mutant background, *KEAP1* inactivation correlates with increased expression of genes involved in glutamine metabolism and the TCA. In this situation, *KRAS/STK11* mutant cells become addicted to glutamine metabolism to boost cell survival [142]. 

## 4. Emerging Altered Cancer Pathways Associated with Immunosuppression in NSCLC

In the last decade, several deregulated cancer pathways associated with immunosuppression have emerged. Hence, targeting these aberrant pathways could represent a strategy to overcome resistance to immunotherapy treatment. Some of these altered pathways are caused by genetic alterations (such as gene mutation and amplification), while others seem to be mainly related to transcriptomic changes. 

### 4.1. Wnt/β-Catenin

Alteration of the Wnt/β-catenin pathway is a notable potential mechanism of immune escape and resistance to ICIs [143]. Expression of β-catenin is implicated in the reduction of TILs in NSCLC patients, even when these tumors have a high TMB load [144]. In this study, the infiltration of CD8+ and CD11c+ cells into the tumor was significantly lower in β-catenin-positive cases compared to that in negative β-catenin cases. The β-catenin-positive group had a shorter OS compared to that in the negative group. 

### 4.2. YES1

v-Yes1 Yamaguchi sarcoma viral oncogene homolog 1 (YES1) is a non-receptor tyrosine kinase that belongs to the SRC family of kinases (SFKs). This family plays an important role in the activation of prosurvival signaling pathways, migration and invasion [145]. Many studies have evidenced the protumorigenic activity of YES1 in a wide variety of solid tumors, such as breast cancer [146], prostate cancer [147], gastric cancer [148], liver cancer [149], NSCLC [150] and SCLC [151], among others. In NSCLC, *YES1* gene amplification was found in 15% of LUADs and 25% of LUSCs, with a significant positive correlation between YES1 gene copy number and mRNA expression [150]. YES1 phosphorylates FAK and YAP1 [145], both involved in the promotion of an immunosuppressive TME characterized by higher infiltration of Tregs and decreased numbers of cytotoxic T cells [152,153] Targeting SFKs with the FDA-approved inhibitor, Dasatinib, not only reduced tumor growth in preclinical models, but also acted as an immunomodulatory drug by promoting a T-cell-inflamed TME. In NSCLC models, Redin et al. have demonstrated a synergistic antitumor effect between dasatinib and anti-PD-1, with a reduction in the number of Tregs and an increase in CD4+ and CD8+ T cells. More interestingly, the authors demonstrated that genetic inhibition of *YES1* also potentiated the response to anti-PD-1, suggesting that YES1 could be a key SFK involved in immunosuppression [154].

### 4.3. DSTYK

Another example of a novel oncogene with immunosuppressive properties is the dual cytoplasmic tyrosine kinase DSTYK, which is amplified in 7% of NSCLC patients. Genetic amplification of *DSTYK* sustains mTOR-mediated autophagy dependency in tumor cells. This pathway activation is directly related to NF-kB pathway maintenance, making tumor cells resistant to TNF-α–mediated CD8+ killing. In this way, by altering the autophagy pathway mediated by mTOR and its direct link to the TNFR1 pathway, the inhibition of DSTYK in lung cancer cells sensitizes tumors to CD8-mediated attack [155].

### 4.4. MUC1 and Hippo Pathways

Mucin 1 (MUC1) is a protein of the plasma membrane that is highly expressed in NSCLC. The MUC1-C variant induces PD-L1 expression and represses IFN-γ [156]. Targeting MUC1-C in NSCLC reduces levels of PD-L1 and activates effector cells of innate and adaptive immunity [156]. 

The Hippo pathway regulates cell proliferation, apoptosis and self-renewal. YAP and its partner TAZ remain in the cytosol and are finally degraded when the pathway is “on”. When the pathway is “off”, the YAP/TAZ complex translocates to the nucleus and exerts protumor effects, including reduction of the immune response through PD-L1 expression [157,158]. 

### 4.5. DNA Damage Repair (DDR)

Pathways related to DNA damage repair (DDR) have also been explored in NSCLC. Li et al. evaluated signatures related to DDR alterations and prognosis in NSCLC, and identified a prognostic model comprising a DDR-gene set [159]. NSCLC patients were then classified according to this model into high- or low-risk groups, each with differences in immune cells and clinical outcomes. High-risk patients showed a lower number of DDR mutations and a worse prognosis but responded better to immunotherapy. On the contrary, low-risk patients showed a tendency for a better outcome, but they had a lower response to ICIs. However, patients were more sensitive to DNA-damaging chemotherapy. 

High TMB familial cancer has been associated with improved survival upon treatment with ICIs in patients with NSCLC, regardless of the status of genes involved in DNA damage [160]. Chen et al. described that co-mutation of *TP53* and *ATM* occurs in a subgroup of NSCLC patients with an increased TMB and response to ICIs [161]. 

## 5. Cancer Cell-Intrinsic Genetic Alterations That Contribute to Immunosuppression in SCLC

### 5.1. Main Genetic Alterations Driving Immunosuppression in SCLC

In contrast to NSCLC, SCLC is characterized by the almost universal loss (90–100%) of *TP53* and *RB1*, both TSG regulators of cell cycle and proliferation [6]. *TP53* and *RB1* mutations cause genomic instability, which, together with the high tobacco-induced TMB found in SCLC tumors, would suggest that SCLC is a highly immunogenic neoplasm [162]. However, SCLC displays a suppressive phenotype, with immunotherapy responses lower than expected based on TMB. Several cancer cell-intrinsic and cell-extrinsic mechanisms have been associated with inherent resistance. Among the tumor cell-related factors, *RB1* loss has been linked to an accumulation of MDSC [162,163]. In addition, low or no MHC-I or PD-L1 expression in tumor cells has been reported, resulting in limited efficacy of ICIs [164,165]. Tumor PD-L1 is found in less than 1% of cells and does not act as a predictor of response to ICIs in SCLC, unlike in NSCLC [162,166]. Moreover, downregulation of MHC-I is present in almost 70% of SCLC patients, which is critical in immunosuppression, as the lack of antigen presentation impedes T cell activation [164]. MHC-I downregulation is generally due to lysine-specific demethylase 1a (LSD1)-driven epigenetic silencing. Some studies have suggested LSD1 as a regulator of MHC-I in SCLC, whose inhibition restores MHC-I expression, activates IFN signaling and induces immune activation, therefore reversing the resistance to immunotherapy. An additional combination of LSD1 inhibition and ICIs has shown an enhancement in the antitumor response in SCLC. In particular, a study showed that the LSD1 inhibitor bomedemstatin combined with PD-1 blockade promoted a hot TME by increasing CD8+ T cell infiltration, thus significantly reducing tumor growth in a syngeneic model of SCLC [167,168] (Figure 2).

TMB has been correlated in SCLC with a better response to nivolumab alone or in combination with ipilimumab, based on a whole exome sequencing characterization performed in a cohort of 201 patients from the Checkmate 032 clinical trial [166]. Nevertheless, assessing TMB in SCLC tumors is very challenging due to the lack of tissue resections or biopsies, because diagnosis usually takes place at metastatic stages [169,170,171].

The relationship between gene alterations and the SCLC TME needs thorough characterization. As mentioned previously, inactivation of *TP53* and *RB1* can be considered almost universal in SCLC. The impact of other mutations/gene amplifications found in SCLC, such as *PTEN*, *CREBBP*, genes belonging to the Notch pathway, *MYC*, *MYCL*, *MYCN*, *FGFR1* or *GNAS* on the TME features has not been reported yet. Some of these genetic alterations have been associated with resistance to immunotherapy in other cancer types, including *CREBBP* [172] and *PTEN* [82,173], and it is expected that they can also play a similar role in SCLC. 

### 5.2. Molecular Subtypes

Despite most SCLC tumors sharing many genetic alterations, the following two molecular subgroups can be found: neuroendocrine (NE) tumors (~70%), which express neuroendocrine markers such as synaptophysin, gastrin-releasing peptide, chromogranin and neural cell adhesion molecule-1, and non-NE tumors. NE-SCLC neoplasms include tumors expressing the transcription factor ASCL1 (SCLC-A) or NEUROD1 (SCLC-N), whereas non-NE tumors express POU2F3 (SCLC-P) or YAP1 (SCLC-Y) [174]. Low-NE SCLCs show increased immune cell infiltration and correlate with a higher benefit from ICIs in comparison with NE-high tumors [163]. In fact, patients with SCLC-Y tumors show a better prognosis and increased immune cell infiltrates, with a higher number of INF-γ-related genes as well as HLA expression and T-cell receptors (*Tcra*, *Tcrb*, *Tcrg*, *Tcrd*) than the other groups [174,175]. Recently, Gay et al. described a novel non-NE SCLC subtype called inflamed “SCLC-I”. This new SCLC-I subset was defined by an inflamed gene signature that included immune checkpoint molecules (PD-L1, PD-1, CTLA-4, TIGIT, VISTA, ICOS, LAG3), genes encoding HLAs, as well as the chemokines CCL5 and CXCL10. In addition, immune cell infiltration (CD8+ T cells, macrophages, B lymphocytes and NK cells) was higher in SCLC-I tumors than in the rest of the subtypes [176]. These results were also validated at the single-cell level by Tial et al., who revealed that NE-low tumors showed increased immune cell infiltration and inflammatory gene signatures [177].

### 5.3. Notch Signaling

The Notch signaling pathway has been described as having an important function in the heterogeneity of SCLC. It is involved in cell differentiation, proliferation, survival and apoptosis, and exhibits tumor suppressive functions in SCLC [6,178]. Notch acts as a tumor suppressor and is inactivated in the majority of SCLC cases that negatively regulate NE differentiation. Its repression is driven by mutations in genes of the Notch pathway or by expression of the Notch inhibitor ligands DLK1 and DDL3 [6,179]. Interestingly, the Notch signaling pathway is implicated in the regulation of some components of the TME, playing a key role in the activation of CD8+ and CD4+ T cells [180] and in the differentiation of Tregs, NKs and DCs [181]. In support of this, patients with activated Notch pathways have shown clinical benefit from ICIs in relapsed SCLC [182]. Up to 85% of SCLC patients showing DLL3 tumoral expression can benefit from Rovalpituzumabtesirine (Rova-T) (Figure 2). This targeted therapy has been used in a phase II clinical trial in SCLC patients, and, despite showing a median survival of 5.7 months and a 19% response rate, a high proportion of patients (40%) developed > grade 3 toxicities. The combination with anti-PD-1 therapy synergistically reduced tumor growth and increased T cell infiltration, DC activation and pro-inflammatory cytokines related to the recruitment of T lymphocytes, in a murine SCLC model [183]. Moreover, Chen et al. observed that targeting DLL3 with a bispecific antibody resulted in suppression of tumor growth in vivo, which was more pronounced when combined with anti-PD-1 inhibition. SCLC patients could benefit from ICIs with this combination [184].

### 5.4. Targeting DNA Damage Repair (DDR) in SCLC to Reshape the TME

There is an emerging interest in the study of proteins related to DDR, such as ATM, ATR, PARP or CHEK1, in SCLC. The DDR pathway is hyperactivated in SCLC, and DDR-targeting drugs increase neoantigen production and immunogenicity in SCLC [185,186,187]. Therefore, several authors have explored whether targeting DDR proteins could represent a strategy to reinvigorate the TME and promote responses to ICIs in SCLC [185]. Targeting PARP in combination with immunotherapies exhibited synergistic effects on tumor inhibition [188,189] (Figure 2). Moreover, a recent study showed that the combination of the PARP inhibitor niraparib with radiotherapy activated the cGAS/STING immune response pathway, upregulated PD-L1 tumor cell expression and induced immunogenic cell death. The addition of anti-PD-1 to this dual combination further increased the infiltration of CD3+ and CD8+ T cells and reduced the number of exhausted T cells [190]. Similar results were obtained by Sen et al., showing that targeting DDR was a therapeutic strategy for SCLC. In this study, pharmacological inhibition of DDR not only enhanced expression of PD-L1 in vitro and in vivo in a wide variety of SCLC models, but also increased levels of TILs. Additional combinations with anti-PD-L1 increased both CD8+ and CD3+ T cell infiltration and promoted tumor regression in syngeneic SCLC models. Furthermore, the STING-TBK1-IRF3 pathway was suggested as a driver of the anti-tumor activity when targeting DDR, which increased the levels of the chemokines CXCL10 and CCL5, responsible for the activation and function of cytotoxic T cells [189]. Despite the antitumor growth observed in preclinical SCLC models when targeting PARP plus anti-PD-1, clinical trials combining the PARP inhibitor olaparib with immunotherapy have failed to show clinical benefit. In a phase II clinical trial with relapsed SCLC patients receiving durvalumab and olaparib, only 21% of patients showed meaningful antitumor activity, and the overall response rate (ORR) was similar to that observed in a clinical trial with nivolumab alone [191].

### 5.5. Other Targets to Reshape the SCLC TME

Inhibition of CDK7, a master regulator of cell-cycle progression, has shown promising results in SCLC models. CDK7 blockade impairs DNA replication and cell cycle, causing replicative stress and, therefore, activating immune response signaling that can be potentially improved with ICIs (Figure 2). The CDK7-inhibitor YKL-5-124 increased pro-inflammatory cytokines and chemokine production (INF-α, TNF-α, CXCL10) in vitro. Combinations with anti-PD-1 induced an increase in the number of CD4+ T cells and tumor-resident DCs in vivo [192].

Evaluation of novel inhibitory checkpoint molecules beyond the PD-1/PD-L1 axis has nominated B7-H6 as a potential immune-related marker. Thomas et al. observed that high expression of B7-H6 correlated with better PFS in SCLC, increased CD8+ T cell immune infiltration and lowered the number of activated NK cells [193]. Weiskopf et al. found CD47 to be highly expressed in SCLC. This surface molecule binds to signal-regulatory protein alpha (SIRPα), inhibiting macrophages and leading to immune evasion. This study demonstrated that targeting CD47 reduced tumor growth in vivo by restoring macrophage-induced phagocytosis [194].

## 6. Epigenetic Changes That Modify the TME in Lung Cancer and Immunotherapy Response

### 6.1. Epigenetic Alterations in NSCLC and SCLC

Apart from genetic changes, epigenetic alterations play a crucial role in carcinogenesis, tumor progression and metastasis. In fact, epigenetic modulation has been proposed as an effective therapeutic strategy to promote tumor immunogenicity and clinical benefit from ICIs [195]. The epigenetic regulation of gene expression happens at different levels: (a) protein (histone modifications: histone deacetylases (HDACs), histone acetyltransferases (HATs), histone methyltransferases (HMTs), histone demethylases (HDMs)); (b) DNA (DNA methylation: DNA demethylases and DNA methyltransferases (DNMTs)); and (c) non-coding RNA (ncRNAs) [196,197]. In general, DNA methylation is the best studied epigenetic mechanism, being responsible for gene silencing. These modifications have a huge impact not only on cancer hallmarks (e.g., proliferation, invasion, apoptosis, regulation of cell cycle), but also on signaling pathways [197].

Epigenetic alterations are highly relevant in both NSCLC and SCLC. In the case of SCLC, epigenetic changes are more frequent than genetic aberrations and contribute to tumor aggressiveness [198]. Alterations of the epigenome and their relationship with the TME are wide and complex and cannot be sufficiently summarized here. We provide some reported data that exemplify the relevance of this process in cancer (mainly in NSCLC and SCLC), but other reviews have addressed this topic in more depth [195,199,200].

### 6.2. Epigenetic Changes in Lung Cancer Related to the Immune TME

Epigenetic changes can alter the TME through different mechanisms, including downregulation of proteins involved in the antigen presentation machinery, changes in the expression of immune checkpoints and modification of the transcriptomic program of tumor-infiltrating immune cells. 

DNA methylation has been described as promoting T-cell exhaustion. In a study carried out by Wang et al., inhibiting DNA methylation in combination with PD-1 blockade enhanced the rejuvenation of CD8+ T cells [201]. CCL2 is known to modulate the immunosuppressive state in tumors, and inhibition of its expression with DNMT-targeting drugs favors a more pro-inflammatory TME, reduces the number of Tregs and increases CD8+ T cells in lung cancer [202]. Moreover, DNA methylation inhibition is being studied as a mechanism to prevent exhaustion, favoring the efficacy of anti-PD-1 antibodies [203]. 

Another mechanism of resistance to ICIs is poor tumor T-cell infiltration. It has been demonstrated that the chemokines CXCL10 and CCL5 are linked to better response to immunotherapy, high T-cell infiltration and patient survival [204]. Zhen et al. developed a strategy to augment the expression of T-cell chemokines and T-cell tumor infiltration in order to enhance the response to immunotherapy. For this purpose, they tested a variety of FDA-approved agents that induce T-cell chemokine expression. HDAC inhibitors were able to increase the number of TILs and macrophages. Targeting HDAC not only showed antitumor efficacy in vivo, but also potentiated the response to PD-1 therapy in many lung cancer models [205]. In correlation with this, some studies have elucidated the potential function of HDAC inhibitors in inducing MHC-I/II molecules and co-stimulatory molecules, including CD40 and CD80 [206,207]. Similar results were obtained by Briere et al., demonstrating that mocetinostat, a class I/IV HDAC inhibitor, upregulated antigen presentation genes (HLA) and PD-L1 expression in many NSCLC cell lines in vitro. Moreover, an increase in intratumoral CD8+ T cells and a decrease in Treg and MDSC populations were observed in vivo. As a consequence, mocetinostat potentiated the efficacy of PD-L1 blockade [208].

As described in the literature, HDAC inhibitors exhibit direct antitumor effects as well as immunomodulatory effects by increasing the expression of pro-apoptotic genes related to death receptor pathways (such as TRAIL and DR5) or intrinsic apoptotic pathways (Bax, Bak and APAF1), in addition to decreasing the expression of pro-survival genes (BCL-2 and XIAP). Moreover, HDAC inhibitors lead to upregulation of costimulatory/adhesion molecules (CD80, CD86, HLA), intracellular adhesion molecule-1 (ICAM) and MHC-I/II proteins [209].

In advanced NSCLC, the HDAC inhibitor vorinostat enhanced the response to Carboplatin and Paclitaxel, but without additional survival benefit. Moreover, in advanced chemo-refractory NSCLC, entinostat plus erlotinib increased survival in a small group of patients presenting high tumor E-cadherin levels [210]. Many studies are focusing on the combination of DNMT and HDAC inhibitors in order to reactivate TSG, as demonstrated in vitro by Boivin et al. in NSCLC cell lines [211]. As epigenetic modifications tend to increase tumor antigens and immunogenicity, the combination of HDAC blockers with ICIs has been proposed as a promising strategy to treat solid tumors.

Epigenetic changes have also been used as indicators of ICI response. Duruisseaux et al. studied the correlation between epigenetic features and clinical benefit of NSCLC patients treated with PD-1 blockade. They found an epigenomic signature (EPIMMUNE) that was associated with improved PFS and OS [212]. However, the EPIMMUNE-positive signature was not associated with PD-L1 expression, presence of CD8+ lymphocytes or TMB [212]. To test if changes in global methylation influenced the clinical benefit of ICIs, Jung et al. studied the methylome and exome of tumor specimens from NSCLC patients treated with anti-PD-1/PD-L1. They found that global methylation was significantly correlated with poor clinical responses [213]. 

A common mechanism of ICI resistance in SCLC includes the loss of MHC-I [81]. Nguyen et al. elucidated a strategy to restore MHC-I expression in SCLC by inhibition of lysine-specific demethylase 1 (LSD1), which promotes immune activation and antitumor response to ICIs [167]. It is generally accepted that MHC-I loss is mainly due to epigenetic silencing. Thus, an epigenetic regulator that might increase MHC-I expression could increase immunotherapy efficacy. EZH2 inhibitors upregulate MHC-I expression, as shown in a variety of murine models including SCLC [214]. The majority of SCLC cases show EZH2 overexpression, leading to downregulation of the TGF-β-SMAD pathway and upregulation of ASCL1 and tumor progression [215]. 

Beyond tumor cells, epigenetic modifications have been described as having a key role in the TME through metabolic reprogramming of immune cell populations, including activation of anti-tumor T effector lymphocytes and APCs, as well as inhibition of immunosuppressive MDSC and Tregs. This topic has been thoroughly reviewed by Dai et al. [196]. 

Clinical trials using epigenetic modulators and immunotherapy (in combination or not with chemotherapy) are underway. Some of these relevant trials are shown in Table 1.

## 7. Personalized Immunotherapy beyond ICIs

Current approved immunotherapy for lung cancer basically relies on the administration of anti-PD-1 or anti-PD-L1 antibodies, and the only personalized factor is based on tumor expression of PD-L1 and TMB as possible indicators of response. To make personalized treatments a closer reality, the definition of how the different mutations and transcriptomic alterations of each tumor may reshape the TME is critical. Expression of alternative immune checkpoints will also be essential to determine whether other ICIs may have a clinical impact on patients with these immunophenotypes. Many of these ICIs, alone or in combination, are still under investigation to determine if clinical benefits can be extended beyond the effects of anti-PD-1/PD-L1 [14]. 

Other immunotherapy approaches included vaccines, albeit with very limited success. The main therapeutic attempts using vaccines in NSCLC were the MAGRIT, START and STOP clinical trials, none of which reached the primary endpoints [216]. The MAGRIT trial tested vaccination with melanoma-associated antigen 3 (MAGE-A3), whereas the target for START was MUC1. In the STOP trial, a mixture of allogenic tumor cells with a TGF-β2 antisense molecule was used. 

Chimeric antigen receptor (CAR) cells are genetically engineered T cells (or other immune cells) that specifically recognize and bind tumor antigens [217]. They show great promise for NSCLC treatment, although several important issues have hindered their application to solid tumors so far: (1) Lack of specific tumor antigens for solid tumors; the existing CAR-T cells use proteins overexpressed in tumors that are found in other healthy tissues as well (on-target/off-tissue), which may lead to severe toxicity; (2) the presence of an immunosuppressed TME that may hinder the accessibility of CAR-T cells to the target; (3) tumor antigen escape; (4) toxicity associated with the therapy, including neurotoxicity and cytokine release syndrome (CRS). Several CAR-T cell trials are investigating antitumor response and toxicity in NSCLC, most of them in very early phases. The most commonly targeted antigens have been EGFR, mesothelin (MSLN), MUC1, PD-L1 and CEA. More information about this issue can be found in the review paper published by Qu et al. [218]. In addition to CAR-T cells, exploration of the potential of CAR-macrophages (CAR-M) in cancer seems particularly intriguing, due to the plasticity of these cells in adapting to the tumor niche [219]. Although experiments in NSLC are still lacking, preclinical data on breast, ovarian and pancreatic cancers suggest great potential to fight the immunosuppressed TME. CAR-Ms could overcome many of the challenges that block the activity of CAR-T cells, as they were able to survive in immunosuppressed niches and produce an array of inflammatory cytokines. One example of this approach was to transduce CD14+ monocyte-derived macrophages with an anti-HER-2 CAR, which was effective against ovarian tumors [220]. Results from these therapies are too preliminary to tell if they will be successful in clinical trials for the treatment of NSCLC, but they could represent another breakthrough in personalized immunotherapy. 

## 8. Conclusions

Resistance to immunotherapy in lung cancer is determined by both cancer cell-intrinsic and extrinsic mechanisms. In cancer cells, certain genetic, epigenetic, transcriptomic and metabolic alterations determine the type of TME, which could be associated with sensitivity or refractoriness to ICIs. From these many alterations, mutations in the dominant cancer driver in each tumor are key to configuring the TME, although co-mutations in other genes can substantially modify the response to these agents. In NSCLC, mutations in *KRAS*, *SKT11*(*LKB1*), *KEAP1* and *TP53* and combined mutations of these genes have been found as determinants of immunotherapy response. Currently, there is intense research on how *KRAS* mutations in particular, in conjunction with other genetic alterations, may influence tumor infiltration of immune cell populations. Other pathways where there is a lot of expectation for immunotherapy interventions are those related to DNA damage repair (DDR), in the hope that drugs targeting DDR will increase neoantigen load and immune response. This approach could be particularly relevant for SCLC, although clinical trials have not produced conclusive results so far. 

It is likely that in a few years, our knowledge about how cancer cell alterations determine the TME will allow us to select patients who can benefit from ICIs in a more precise way. A better understanding of each individual TME will also help identifying biomarkers that can improve the predictive value of PD-1 and TMB in lung cancer. Currently, there are numerous in situ high-throughput methods to accurately assess the TME, including multiplex immunofluorescence, nanostring-based digital spatial profiling (DSP), spatial transcriptomics, image mass cytometry, etc. [221]. An important issue for patients who initially respond to ICIs but later develop acquired resistance is determining possible new actionable immune checkpoints in the TME. This would require in situ analysis of a post-treatment biopsy, a practice that is not routinely performed. Although this would constitute a step ahead in personalized medicine, clinical trials need to demonstrate the benefit of using ICIs other than anti-PD-1/PD-L1/CTLA-4 and biomarkers of response. In conclusion, a better understanding of cancer cell-intrinsic changes that associate with a specific TME will translate into the identification of biomarkers of response to ICIs and novel therapeutic strategies. 

## Figures and Tables

**Figure 1 cancers-15-03076-f001:**
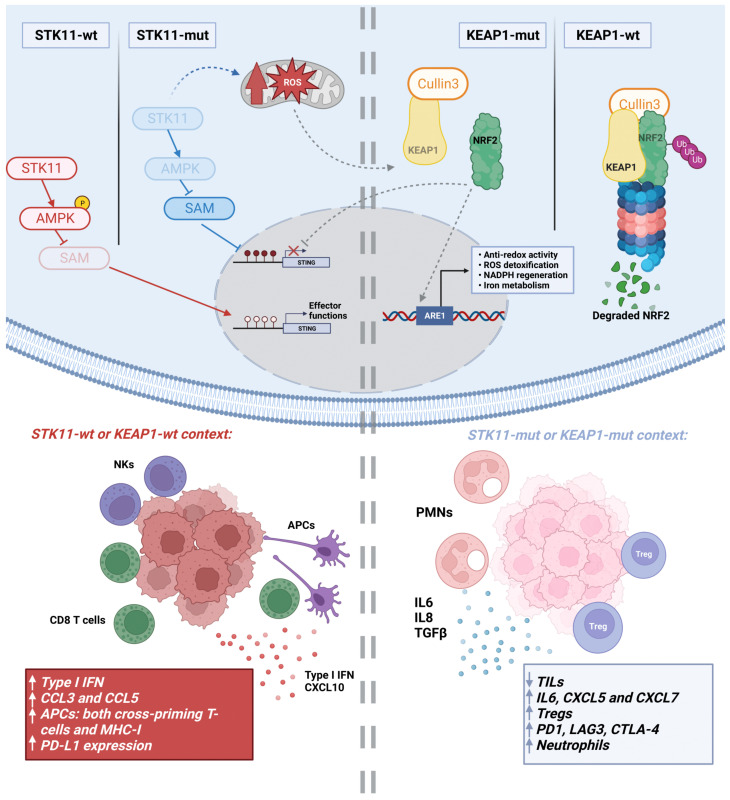
Effect of *STK11* and *KEAP1* mutations on some key cancer cell intracellular pathways and the TME, compared to wild-type tumors. The *STK11* mutation blocks phosphorylation of AMP-activated protein kinase (AMPK) and promotes serine utilization and synthesis of S-adenosyl methionine (SAM), which is a substrate for several epigenetic silencing enzymes, like DNMT1 and EZH2. This, in turn, causes the silencing of STING (STimulator of INterferon Genes). These and other alterations lead to an immunosuppressive TME. KEAP1 functions as an adaptor protein for Cullin3, an E3 ligase that negatively regulates NRF2 activity. KEAP1 binding to NRF2 causes proteasomal degradation of NRF2. The *KEAP1* mutation allows NRF2 to aberrantly bind antioxidant response elements (ARE) and turn on the transcription of genes related to anti-redox activity, ROS detoxification, NADPH regeneration and iron metabolism. NRF2 hyperactivity also alters numerous metabolic-related pathways that, ultimately, affect the activity of immune cells within the TME and cause a lack of response to ICIs. Created with BioRender.

**Figure 2 cancers-15-03076-f002:**
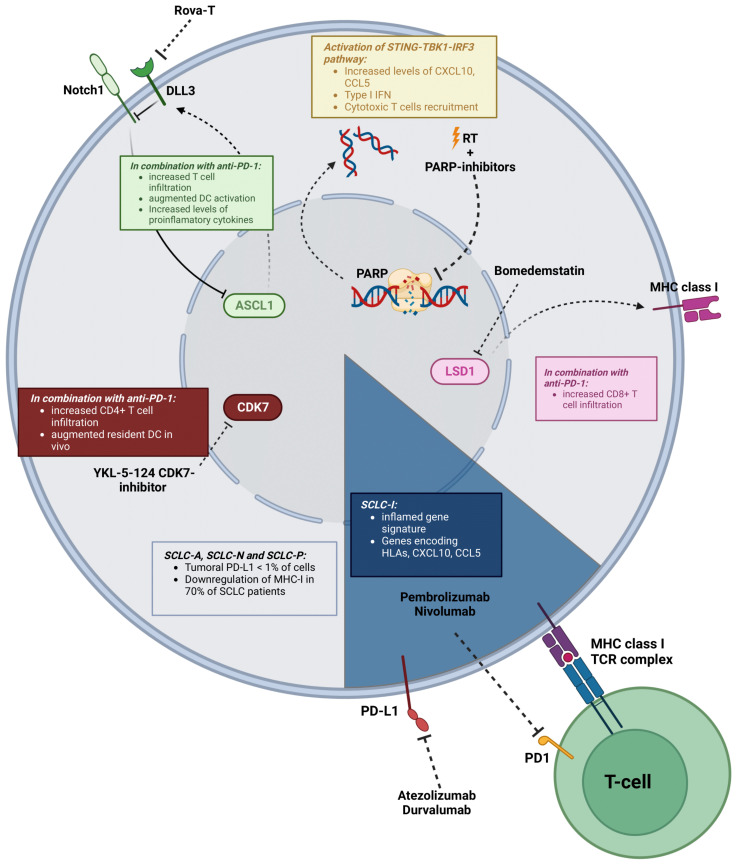
Some of the key modulators of the TME in SCLC and possible therapeutic strategies to overcome resistance to ICIs. SCLC subtypes are characterized by expression of ASCL1 (SCLC-A), NEUROD1 (SCLC-N), POU2F3 (SCLC-P) or YAP1 (SCLC-Y). A novel subtype, SCLC-I, was defined by an inflamed gene signature that included immune checkpoint molecules (PD-L1, PD-1, CTLA-4, TIGIT, VISTA, ICOS, LAG3), genes encoding HLAs and chemokines CCL5 and CXCL10. These tumors are particularly sensitive to anti-PD-1 therapy. In the other subtypes, which are in general refractory to immunotherapy alone, different strategies have been proposed to enhance immunogenicity: (a) rovalpituzumabtesirine (Rova-T) to block the Notch ligand DLL3; (b) use of DNA damage agents, such as radiotherapy or PARP inhibitors, which increase levels of CXCL10, CCL5 and Type I IFN, and recruit cytotoxic T cells; (c) increase the expression of MHC-I, which may be downregulated by lysine-specific demethylase 1a (LSD1)-driven epigenetic silencing. Inhibition of LSD1 (for instance, with bomedemstatin) restores MHC-I expression, activates IFN signaling and induces immune activation; (d) CDK7 is a master regulator of cell-cycle progression in SCLC. CDK7 blockade impairs DNA replication and the cell cycle, causing replicative stress and activation of an immune response. The CDK7-inhibitor YKL-5-124 increases pro-inflammatory cytokines and chemokine production. Created with BioRender.

**Table 1 cancers-15-03076-t001:** Summary of clinical trials of epigenetic therapy alone or in combination with immunotherapy or immunochemotherapy in lung cancer patients. Clinical trial information was obtained from clinicaltrials.gov. Abbreviations: DNMT1 (DNA methyltransferase 1); HDAC (histone deacetylase); CDA (cytidine deaminase); m^5^C (5-methylcytosine); FdCyd (5-fluoro-2′-deoxycytidine); TILs (tumor infiltrating lymphocytes); BET (bromodomain extra-terminal); LSD1 (lysine-specific demethylase 1).

Subtype	Clinical Trial Identifier	Phase and Status	Epigenetic Agent	Target	Combined Agent
NSCLC	NCT01928576	Phase II; Ongoing	Azacitidine Entinostat	DNMT1 HDAC	Nivolumab
NSCLC	NCT03233724	Phase I/II; Completed	Decitabine Tetrahydrouridine	DNMT1 CDA	Pembrolizumab
NSCLC	NCT03220477	Phase I; Ongoing	Guadecitabine Mocetinostat	DNMT1	Pembrolizumab
NSCLC	NCT01209520	Pilot study; Completed	Azacitidine	DNMT1	Cisplatin Carboplatin Paclitaxel Vinorelbine Docetaxel Pemetrexed
SCLC NSCLC	NCT02489903	Phase II; Completed	RRx-001	CD47 and SIRP-α	Cisplatin Etoposide Carboplatin Irinotecan Vinorelbine Doxil Gemcitabine Taxane Paclitaxel Nab-paclitaxel Pemetrexed
NSCLC	NCT02959437	Phase I/II; Terminated	Azacitidine INCB057643 INCB059872	DNMT1 BET LSD1	Pembrolizumab Epacadostat
NSCLC	NCT02664181	Phase II; Ongoing	Oral decitabine Tetrahdrouridine	DNMT1 CDA	Nivolumab
NSCLC	NCT02638090	Phase I/II; Ongoing	Vorinostat	HDAC	Pembrolizumab
NSCLC	NCT02437136	Phase IB/II; Ongoing	Entinostat	HDAC	Pembrolizumab
SCLC	NCT02446704	Phase I/II; Ongoing	Temozolomide	m^5^C	Olaparib
SCLC	NCT01638546	Phase II; Completed	Temozolomide	m^5^C	Veliparib
NSCLC	NCT00423150	Phase II; Terminated	Temozolomide	m^5^C	-
SCLC	NCT01222936	Phase II; Completed	LBH589 Panobinostat	HDAC	-
SCLC	NCT02034123	Phase I; Terminated	GSK2879552	LSD1/KMD1A	-
Metastatic solid tumor	NCT05268666	Phase I/II; Ongoing	JBI-802	LSD1/HDAC6	-
SCLC NSCLC	NCT04350463	Phase II; Ongoing	CC-90011	LSD1	Nivolumab
NSCLC	NCT05467748	Phase I/II; Ongoing	Tazemetostat	EZH2	Pembrolizumab
SCLC	NCT03460977	Phase I; Ongoing	PF-06821497	EZH2	-
Lung neoplasms	NCT00978250	Phase II; Completed	FdCyd Tetrahydrouridine	DNMT CDA	-
NSCLC	NCT02546986	Phase II; Ongoing	CC-486	DNMT1	Pembrolizumab
NSCLC	NCT01478685	Phase I; Completed	CC-486	DNMT1	Carboplatin Nab-paclitaxel
NSCLC	NCT02250326	Phase II; Ongoing	CC-486	DNMT1	Durvalumab Nab-paclitaxel
NSCLC	NCT05573035	Phase I; Ongoing	LYL845	TILs	-
NSCLC	NCT05607108	Phase II; Ongoing	ZEN003694	BET	-
NSCLC	NCT01207726	Phase II; Terminated	Azacitidine Entinostat	DNMT1 HDAC	-
SCLC	NCT05191797	Phase I/II; Ongoing	Bomedemstat	LSD1	Atezolizumab
NSCLC	NCT02635061	Phase IB; Ongoing	ACY-241	HDAC6	Nivolumab
NSCLC	NCT01059552	Phase I; Completed	Vorinostat	HDAC	Cisplatin Pemetrexed Radiation 70Gy
NSCLC	NCT00821951	Phase I; Completed	Vorinostat	HDAC	Radiotherapy
NSCLC	NCT02728492	Phase I; Completed	Quisinostat	HDAC1	Paclitaxel Carboplatin Gemcitabine Cisplatin
NSCLC	NCT00667082	Phase I; Completed	Vorinostat	HDAC	NPI-0052 Marizomib
NSCLC	NCT00005093	Phase III; Completed	Tacedinaline	HDAC	Gemcitabine

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
