# Peer review of "Cancer Cell-Intrinsic Alterations Associated with an Immunosuppressive Tumor Microenvironment and Resistance to Immunotherapy in Lung Cancer"

_cancers, 2023, doi:10.3390/cancers15123076_

Round 1

Reviewer 1 Report

The authors provide a concise overview of the current understanding of the factors affecting response to ICIs in lung cancer patients. The authors highlight the importance of cancer cell-intrinsic and extrinsic mechanisms in determining the response to immunotherapy. It is very well suggested that specific genetic, epigenetic, transcriptomic, and metabolic alterations in cancer cells play a crucial role in configuring the TME, which may affect the sensitivity and refractoriness to ICIs.

I recommend for publication of this review article with minor revisions:

1) 5th paragraph in the introduction, Line 1 "During the last decade.... ROS" - Please cite a reference supporting this information.

2) Page 8 - Line 6: Typographical error of Fatty acids should be corrected.

3) The conclusion could have highlighted the need for further research to explore other potential factors that may affect the response to immunotherapy in lung cancer patients.

Author Response

REFEREE 1

The authors provide a concise overview of the current understanding of the factors affecting response to ICIs in lung cancer patients. The authors highlight the importance of cancer cell-intrinsic and extrinsic mechanisms in determining the response to immunotherapy. It is very well suggested that specific genetic, epigenetic, transcriptomic, and metabolic alterations in cancer cells play a crucial role in configuring the TME, which may affect the sensitivity and refractoriness to ICIs.

I recommend for publication of this review article with minor revisions:

1) 5th paragraph in the introduction, Line 1 "During the last decade.... ROS" - Please cite a reference supporting this information.

2) Page 8 - Line 6: Typographical error of Fatty acids should be corrected.

3) The conclusion could have highlighted the need for further research to explore other potential factors that may affect the response to immunotherapy in lung cancer patients.

Response:

We thank the referee for the positive comments about our review paper. We have made the required changes (points 1-3) in the manuscript. Changes have been highlighted in the new text.

Reviewer 2 Report

Summary

In this review article, the authors addresses the main genetic, epigenetic and metabolic changes associated with response to ICIs in lung cancer, including NSCLC and SCLC. They also describe new emerging pathways linked to the establishment of particular TME that can be favorable or unfavorable for immunotherapy response.

Authors should add lines in the document to facilitate the review process.

When the authors talk about "Metabolic rewiring in NSCLC and its influence on the TME", further implementation is required for recent literature update. Several molecular targets of lactate have been recently identified. Actually, the molecular mechanism driving lactate to aggressiveness should be addressed: e.g. lactate has been shown to affect PHD2 activity, there is a lactate receptor that can actually play a role. Plus, it would take some more details about the potential role of lactate in driving "metabolic symbiosis". It is important to discuss the metabolic interaction between oxidative and glycolytic tumor cells.

It has been published elsewhere that TP53 is the most frequently mutated gene across human cancers. Accumulating evidence has shown that mutations of TP53 not only lead to loss-of-function or dominant negative effect, but also promote a gain-of-function. Specifically, gain-of-function mutant p53 promotes cancer cell motility, invasion and metastasis. Could the authors talk about the TP53 loss of function and the activation of molecular pathways associated with epithelial-to-mesenchymal transition? I believe it may make the topic "Main gene mutations that shape the tumor microenvironment in NSCLC" more attractive to readers. 

Topics should be numbered throughout the manuscript. This makes the document easier to read, as well as facilitates the review process.

Regarding therapy prospects, the authors should explore the topic in more depth. Authors should discuss what is new in the literature about the complex relationship between the immune system and cancer and how this can be used for therapeutic purposes. I missed examples with CAR-T and CAR-macrophages. The authors would introduce the importance of these new technologies in the fight against cancerI believe the topic would be more attractive to readers.

In the topic about "Epigenetic modifications that modify the TME in lung cancer and immunotherapy response", I suggest the authors add a table to illustrate how epigenetic modulation has been proposed as an effective therapeutic strategy to promote tumor immunogenicity and clinical benefit.

Figure legend for Fig.2 should be  rewritten. The figure brings a lot of information, and the caption is poor in content.

Author Response

REFEREE 2

In this review article, the authors addresses the main genetic, epigenetic and metabolic changes associated with response to ICIs in lung cancer, including NSCLC and SCLC. They also describe new emerging pathways linked to the establishment of particular TME that can be favorable or unfavorable for immunotherapy response.

Authors should add lines in the document to facilitate the review process.

When the authors talk about "Metabolic rewiring in NSCLC and its influence on the TME", further implementation is required for recent literature update. Several molecular targets of lactate have been recently identified. Actually, the molecular mechanism driving lactate to aggressiveness should be addressed: e.g. lactate has been shown to affect PHD2 activity, there is a lactate receptor that can actually play a role. Plus, it would take some more details about the potential role of lactate in driving "metabolic symbiosis". It is important to discuss the metabolic interaction between oxidative and glycolytic tumor cells.

Response:

A whole section on lactate (see now 3.1) and its role in the TME has been added to the review: see pages 11 and 12.

It has been published elsewhere that TP53 is the most frequently mutated gene across human cancers. Accumulating evidence has shown that mutations of TP53 not only lead to loss-of-function or dominant negative effect, but also promote a gain-of-function. Specifically, gain-of-function mutant p53 promotes cancer cell motility, invasion and metastasis. Could the authors talk about the TP53 loss of function and the activation of molecular pathways associated with epithelial-to-mesenchymal transition? I believe it may make the topic "Main gene mutations that shape the tumor microenvironment in NSCLC" more attractive to readers. 

Response:

We have now included information on TP53 gain-of-function related to increased cell motility, invasion, metastasis and immunosuppression (see page 8).

Topics should be numbered throughout the manuscript. This makes the document easier to read, as well as facilitates the review process.

Response:

We thank the referee for this indication, which has made the review more clear and easier to follow. Subheadings and numbers have been added.

Regarding therapy prospects, the authors should explore the topic in more depth. Authors should discuss what is new in the literature about the complex relationship between the immune system and cancer and how this can be used for therapeutic purposes. I missed examples with CAR-T and CAR-macrophages. The authors would introduce the importance of these new technologies in the fight against cancer. I believe the topic would be more attractive to readers.

Response:

A new section has been added to address therapy prospects in personalized immunotherapy, including vaccines, CAR-T cells and CAR-T-macrophages (see pages 25 and 26).

In the topic about "Epigenetic modifications that modify the TME in lung cancer and immunotherapy response", I suggest the authors add a table to illustrate how epigenetic modulation has been proposed as an effective therapeutic strategy to promote tumor immunogenicity and clinical benefit.

Response:

Following the referee’s suggestion, we have added a Table to illustrate how epigenetic modulation has been proposed as an effective therapeutic strategy to promote tumor immunogenicity and clinical benefit.

Figure legend for Fig.2 should be rewritten. The figure brings a lot of information, and the caption is poor in content.

Response:

We have expanded the explanation of Figure legend 2 in order to clarify its content.

Reviewer 3 Report

This review article discusses the immune therapy of lung cancer, including the concepts, principles, mechanisms, and modes of action of treatment. The article mentions the critical role of biomarkers such as PD-1, PD-L1, and CTLA-4 in immune therapy for lung cancer. To improve the effectiveness of immune therapy, researchers need to explore methods to change the immune cell community in the lung cancer microenvironment. The article also focuses on differences in gene expression among lung cancer patients, the possibility of resistance after treatment, and adverse reactions and complications of immune therapy. The article proposes some new research directions and strategies. It also mentions ongoing clinical trials and future research directions, including combining immune therapy with radiation therapy, chemotherapy, and targeted therapy and studying interactions between immune cells in the lung cancer microenvironment. In summary, although immune therapy for lung cancer faces many challenges and limitations, the development prospects of this field are broad, and it is expected to become an important field for future lung cancer treatment. The article is rich in content, and I hope to discuss the following issues with the author for reference, hoping to make the text more perfect and helpful:

 1.       In the last sentence of the abstract, the article mentions providing more personalized drugs for lung cancer patients receiving immune therapy. Still, it does not explicitly mention the strategies of personalized immune therapy. The author may consider adding immune therapy plans based on individualized immune microenvironmental features, such as oncolytic viruses, tumor vaccines, and adoptive immunotherapy (including TIL, CAR-T, TCR-T, CAR-NK, etc.).

 2.       In the section "Epigenetic modifications that modify the TME in lung cancer and immunotherapy response," the author explains how epigenetic modifications can lead to immune therapy resistance. The author may consider summarizing the importance of epigenetics at the beginning of the paragraph. For example, epigenetic changes can affect the expression of immune checkpoints, disrupt antigen presentation, inhibit the migration of T cells to the tumor microenvironment, and affect the production of memory T cells by twisting transcription factors.

 3.       References 19, 29, 58, 59, and 77 in the article are all about treating lung adenocarcinoma, but I did not see references for squamous cell carcinoma. Did the author perform a detailed analysis of the differences in gene expression between different subtypes of lung cancer? For example, have differences between lung adenocarcinoma, squamous cell carcinoma, and large cell carcinoma been considered? These three types of lung cancer have different mutation characteristics and gene expression patterns, as well as different clinical features, which can affect the choice of treatment and prognosis assessment for lung cancer patients. Accurately identifying the type of lung cancer is important for selecting the best treatment plan and evaluating the patient's prognosis.

4.       In addition to drug resistance, the author could also consider exploring specific complications and adverse reactions during the immunotherapy process, such as immune-related pneumonitis, rash, inflammatory bowel disease, thyroiditis, liver dysfunction, and neuropathy. Understanding the mechanisms and predictive factors of these adverse reactions can provide more basis for the safe application of lung cancer immunotherapy.

 5.       The article mentions the stage of cancer cell escape. The author could consider whether tumor cells will produce new immune escape mechanisms after lung cancer immunotherapy, leading to decreased treatment efficacy. Many patients experience further tumor progression after receiving immunotherapy. It may be because tumor cells have developed new immune escape mechanisms, such as regulating cell surface markers to avoid immune system attack. Tumor cells can also evade immune attacks by regulating the function of immune cells.

 6.       It is an extremely comprehensive review article with a large number of references, but the structure of the article can be improved by adding more subheadings based on the content of each paragraph to facilitate readers' quick navigation and reading. For example, in the section "Main gene mutations that shape the tumor microenvironment in NSCLC," the author used 14 paragraphs to discuss this topic. Still, it could be improved by adding a subheading every three to four paragraphs based on the content.

 7.       Personalized medicine has been mentioned earlier, and I suggest that the author briefly discusses the importance of personalized medicine in lung cancer immunotherapy again in the discussion section, especially in new methods for determining the tumor microenvironment (TME) and predicting immunotherapy response, such as predicting immune cell infiltration and tumor burden. Through these methods, lung cancer patient's response to immunotherapy can be more accurately predicted, thereby achieving better-personalized treatment.

Author Response

REFEREE 3

This review article discusses the immune therapy of lung cancer, including the concepts, principles, mechanisms, and modes of action of treatment. The article mentions the critical role of biomarkers such as PD-1, PD-L1, and CTLA-4 in immune therapy for lung cancer. To improve the effectiveness of immune therapy, researchers need to explore methods to change the immune cell community in the lung cancer microenvironment. The article also focuses on differences in gene expression among lung cancer patients, the possibility of resistance after treatment, and adverse reactions and complications of immune therapy. The article proposes some new research directions and strategies. It also mentions ongoing clinical trials and future research directions, including combining immune therapy with radiation therapy, chemotherapy, and targeted therapy and studying interactions between immune cells in the lung cancer microenvironment. In summary, although immune therapy for lung cancer faces many challenges and limitations, the development prospects of this field are broad, and it is expected to become an important field for future lung cancer treatment. The article is rich in content, and I hope to discuss the following issues with the author for reference, hoping to make the text more perfect and helpful:

-In the last sentence of the abstract, the article mentions providing more personalized drugs for lung cancer patients receiving immune therapy. Still, it does not explicitly mention the strategies of personalized immune therapy. The author may consider adding immune therapy plans based on individualized immune microenvironmental features, such as oncolytic viruses, tumor vaccines, and adoptive immunotherapy (including TIL, CAR-T, TCR-T, CAR-NK, etc.).

Response:

As discussed for referee 2, we have now included a section on personalized immunotherapy, including vaccines, CAR-T cells and CAR-T-macrophages (see pages 25 and 26). Also, a sentence has been added in the Abstract.

In the section "Epigenetic modifications that modify the TME in lung cancer and immunotherapy response," the author explains how epigenetic modifications can lead to immune therapy resistance. The author may consider summarizing the importance of epigenetics at the beginning of the paragraph. For example, epigenetic changes can affect the expression of immune checkpoints, disrupt antigen presentation, inhibit the migration of T cells to the tumor microenvironment, and affect the production of memory T cells by twisting transcription factors.

Response:

Following the referee’s suggestion, we have added an introductory sentence to stress the importance of epigenetics on the tumor microenvironment (see page 23).

References 19, 29, 58, 59, and 77 in the article are all about treating lung adenocarcinoma, but I did not see references for squamous cell carcinoma. Did the author perform a detailed analysis of the differences in gene expression between different subtypes of lung cancer? For example, have differences between lung adenocarcinoma, squamous cell carcinoma, and large cell carcinoma been considered? These three types of lung cancer have different mutation characteristics and gene expression patterns, as well as different clinical features, which can affect the choice of treatment and prognosis assessment for lung cancer patients. Accurately identifying the type of lung cancer is important for selecting the best treatment plan and evaluating the patient's prognosis.

Response:

In the clinical trials performed so far to test the different immunotherapy regimes, both lung adenocarcinomas (LUAD) and lung squamous cell carcinomas (LUSC) were included, sometimes selecting the patients based on PD-L1 expression and/or TMB. Studies based on mutations in NSCLC patients with that particular genotype, sometimes include both histological types: i.e., for KEAP1 mutations, found in LUAD and LUSC patients; whereas in other cases (i.e. KRAS) only LUAD were included. In the review, we have tried to explain the TME based on mutations, taking into account that they can be found in both histological types or in LUAD. For LUSC there is no conclusive clinical information about specific mutations like FGFR or PIK3CA in this histological type and the associated TME and response to immunotherapy.

In addition to drug resistance, the author could also consider exploring specific complications and adverse reactions during the immunotherapy process, such as immune-related pneumonitis, rash, inflammatory bowel disease, thyroiditis, liver dysfunction, and neuropathy. Understanding the mechanisms and predictive factors of these adverse reactions can provide more basis for the safe application of lung cancer immunotherapy. 

Response:

This is an important issue for lung cancer patients treated with immunotherapy, but we have not found relevant information about how specific mutation patterns or genetic alterations may be associated with particular side effects that may not be found in other patients. In the studies where specific mutations were studied in NSCLC patients, no description about particular side effects were reported.

The article mentions the stage of cancer cell escape. The author could consider whether tumor cells will produce new immune escape mechanisms after lung cancer immunotherapy, leading to decreased treatment efficacy. Many patients experience further tumor progression after receiving immunotherapy. It may be because tumor cells have developed new immune escape mechanisms, such as regulating cell surface markers to avoid immune system attack. Tumor cells can also evade immune attacks by regulating the function of immune cells.

Response:

We have now included immune escape mechanisms during tumor evolution and tumor heterogeneity (see pages 4 and 5).

It is an extremely comprehensive review article with a large number of references, but the structure of the article can be improved by adding more subheadings based on the content of each paragraph to facilitate readers' quick navigation and reading. For example, in the section "Main gene mutations that shape the tumor microenvironment in NSCLC," the author used 14 paragraphs to discuss this topic. Still, it could be improved by adding a subheading every three to four paragraphs based on the content.

Response:

We thank the referee for this comment. We have now introduced subheadings to make the reading more friendly.

Personalized medicine has been mentioned earlier, and I suggest that the author briefly discusses the importance of personalized medicine in lung cancer immunotherapy again in the discussion section, especially in new methods for determining the tumor microenvironment (TME) and predicting immunotherapy response, such as predicting immune cell infiltration and tumor burden. Through these methods, lung cancer patient's response to immunotherapy can be more accurately predicted, thereby achieving better-personalized treatment.

Response:

As suggested, we have now briefly discussed the importance of personalized medicine in lung cancer immunotherapy and the usefulness of new immunophenotyping methods for determining the tumor microenvironment (TME) in the conclusion section (see page 27).

Round 2

Reviewer 2 Report

I thank the authors for clarifying my questions. In this new version, all my comments have been properly addressed, and in my opinion the manuscript is suitable for publication.